# Peek-a-Boo: What (More) is Disguised in a Randomly Weighted Neural Network, and How to Find It Efficiently

**Xiaohan Chen[1], Jason Zhang[1,2], Zhangyang Wang[1]**
[1]University of Texas at Austin, [2]Carnegie Mellon University
{xiaohan.chen, jzhang27143, atlaswang}@utexas.edu

## Abstract

Sparse neural networks (NNs) are intensively investigated in literature due to their appeal in saving storage, memory, and computational costs. A recent work (Ramanujan et al., 2020) showed that, different from conventional pruning-and-finetuning pipeline, there exist *hidden subnetworks* in randomly initialized NNs that have good performance without training the weights. However, such "hidden subnetworks" have mediocre performances and require an expensive `edge-popup` algorithm to search for them. In this work, we define an extended class of subnetworks in **randomly initialized NNs** called *disguised subnetworks*, which are not only "hidden" in the random networks but also "disguised" – hence can only be "unmasked" with certain transformations on weights. We argue that the unmasking process plays an important role in enlarging the capacity of the subnetworks and thus grants two major benefits: *(i)* the disguised subnetworks easily outperform the hidden counterparts; *(ii)* the unmasking process helps to relax the quality requirement on the sparse subnetwork mask so that the expensive `edge-popup` algorithm can be replaced with more efficient alternatives. On top of this new concept, we propose a novel two-stage algorithm that plays a *Peek-a-Boo* (**PaB**) game to identify the disguised subnetworks with a combination of two operations: (1) searching efficiently for a subnetwork at *random initialization*; (2) unmasking the disguise by learning to transform the resulting subnetwork's remaining weights. Furthermore, we show that the unmasking process can be efficiently implemented (a) *without referring to any latent weights or scores*; and (b) by only leveraging *approximated gradients*, so that the whole training algorithm is computationally light. Extensive experiments with several large models (ResNet-18, ResNet-50, and WideResNet-28) and datasets (CIFAR-10, CIFAR-100 and ImageNet) demonstrate the competency of PaB over `edge-popup` and other counterparts. Our codes are available at: https://github.com/VITA-Group/Peek-a-Boo.

## 1 Introduction

Recent years have seen substantial efforts devoted to scaling neural networks (NNs) to enormous sizes (He et al., 2016; Devlin et al., 2019). Parameter-counts are frequently measured in billions rather than millions, with the time and financial outlay necessary to train these models growing in concert. *Sparse NNs*, whose large portions of parameters are zero, have been studied to address those gaps, saving storage, memory and computational costs. Conventional approaches first train dense NNs, and then prune the trained NNs to high levels of sparsity (Han et al., 2015; Guo et al., 2016). Those methods significantly reduce the inference complexity yet cost even greater computational resources and memory footprints at training.

An emerging field has explored what roles sparsity can play in neural networks, spawning different directions of research. One of them follows the typical pruning-and-finetuning pipeline in conventional pruning methods, but explored the prospect of directly training smaller, sparse subnetworks

---

*Work done while the author was at the University of Texas at Austin.

in place of the full models without sacrificing performance. One key idea is to reuse the sparsity pattern found through pruning and training a sparse network from scratch. The seminal work of the *lottery ticket hypothesis* (LTH) (Frankle & Carbin, 2019) demonstrated that standard dense NNs contain sparse matching subnetworks (called "winning tickets") capable of training in isolation to full accuracy. In other words, we could have trained smaller networks from the start if only we had known which subnetworks to choose. Additionally, other works also showed that sparsity might even emerge at initialization, before training ever starts and with minimal-to-no dependence on data, such as SNIP (Lee et al., 2019), GraSP (Wang et al., 2020), and SynFlow (Tanaka et al., 2020). Those methods suggest a tantalizing possibility of "pruning-at-initialization" for finding sparse subnetworks without overhead, even though it remains under debate whether their found subnetworks have inferior quality to lottery tickets found with IMP (Frankle et al., 2020). The consensus seems to be that, a "good" sparse topology can be very crucial and informative for successful training.

Another set of efforts (Zhou et al., 2019; Wortsman et al., 2020) suggest that "masking (pruning) is training", e.g., by strategically zeroing out randomly initialized weights, the network can reach far-better-than-chance accuracy. (Ramanujan et al., 2020) found that from a "sufficiently over-parameterized" dense NN with random initialization, there exists a *hidden subnetwork* that achieves competitive accuracy (w.r.t. to the fully trained dense NN), without any training on the remaining weights. This finding was later theoretically justified by (Malach et al., 2020; Pensia et al., 2020; Orseau et al., 2020). Ivan & Florian (2020) tried an alternative training scheme, by only flipping the weight signs in a dense NN while preserving their initialized magnitudes. Sreenivasan et al. (2021) further showed that such hidden subnetworks also exist in networks with binary weights. All aforementioned works suggest that it is possible to only update the connectivity patterns without changing the magnitudes of the weights to achieve decent performance.

However, many of them only tried on rather underlined{small NNs} (Zhou et al., 2019; Ivan & Florian, 2020), with notable accuracy drops compared to full gradient-based training. This is intuitively understandable because NNs' expressiveness will be strictly constrained if pruning is the only allowable operations during training. Under such rigid restrictions, to obtain hidden subnetworks with acceptable performance requires high-quality connectivity patterns with appropriate pruning ratios. Ramanujan et al. (2020) found that the hidden subnetworks with either too high or too low pruning ratios have poor performance. What is even worse is that the tedious optimization needed for identifying the hidden subnetworks. All of those works used some form of latent weights or score functions for all NN weights, which are cast into binary decisions to mask their corresponding weights or not during forward pass and are trained during backpropagation, e.g., the `edge-popup` algorithm in (Ramanujan et al., 2020). Those latent weights or scores make their training even more costly than training standard NNs, and the non-differentiable binarization operation may also inject instability.

## 1.1 OUR CONTRIBUTIONS AND RATIONALES

Inspired by the recent advances in sparse NNs and the philosophy of "masking is training", this paper first proposes the concept of *disguised subnetworks* in **randomly initialized NNs**, as the extension to the "hidden subnetworks" found in (Ramanujan et al., 2020). At its name suggests, a disguised subnetwork is hidden as a sparse subnetwork in a dense NN at its random initialization. This subnetwork, without achieving good performance at the beginning, can have its "disguise" *unmasked* by learning some simple transformations on the remaining weights to lower the training loss. To preserve the randomness in the subnetwork at the best, we only consider transformations that do not alter the magnitudes of the remaining weights, such as shuffling and sign flipping. In this work, we will focus on sign flipping specifically.

We argue that the extra *unmasking* step in disguised subnetworks grants essential advantages over plain hidden subnetworks by significantly enlarging the expressiveness of the subnetworks. Such advantages immediately bring several benefits. Firstly, with much better capacity and expressiveness, disguised subnetworks can easily achieve superior performances compared with hidden subnetworks and thus have better scalability to larger NNs and datasets. Secondly, the searching of the sparse subnetwork and the following unmasking process can compliment each other for better performance, which relaxes the quality requirement on the sparse mask. This benefit enables us to replace the unstable and expensive searching algorithms in previous works with more efficient pruning algorithms. The rationale behind is that only learning transformations on randomly initialized weights cannot create sparsity that a trained NN often shows to possess (Zhou et al., 2019) while only pruning has too limited flexibility which can be drastically boosted by allowing for further weight processing.

Based on this reasoning, we propose a two-phase algorithm called *Peek-a-Boo* (**PaB**), which consists of the *searching* and *unmasking* phases involving two operations: (1) sparsifying a deep neural network with efficient pruning-at-initialization methods at *its random initialization*; (2) learning to flip the signs of the resulting subnetwork's remaining weights to lower the training loss. Note that we **never modify weight magnitudes** after being initialized aside from removing weights. PaB absorbs and improves the idea "masking is training" (Zhou et al., 2019), by unifying the two streams of ideas: pruning (i.e., applying {0,1} masks to weights) and flipping signs (i.e., applying {-1,1} masks). Moreover, for its second stage of unmasking, we optimize weight signs **without extra latent weights or scores** (Helwegen et al., 2019). Such optimization method frees us from using the heavy latent parameters nor the unstable straight-through-estimators (STEs), which were assumed by all previous methods. We further demonstrate that **an even more efficient implementation** could be achieved by using low-cost gradient predictors for approximated gradients (Wang et al., 2019), as PaB might be tolerant to "coarser" gradient information since we only flip signs.

Despite remaining as an empirical exploration, the success of PaB also urges theoretical reflections on whether sparse NNs have to follow the same optimization strategies as the dense ones, or whether they can utilize new optimizers as simple as sign flipping. On one hand, pruning can often cause more difficult optimization landscapes for gradient descents, as training a sparse NN essentially becomes a constrained and non-smooth optimization problem (Evci et al., 2019). On the other hand, assuming a good sparse mask is known *a priori*, the possible solution space has also been significantly reduced, which may suggest the existence of "shortcuts" to reach the optima in this shrunken space, such as the "solution locality" of LTH recently observed by (Liu et al., 2021).

## 2 RELATED WORKS

### 2.1 SPARSE NEURAL NETWORKS

Pruning methods (Hassibi et al., 1993; LeCun et al., 1989) have witnessed great advances to reduce the increasing inference complexities of NNs. Magnitude-based pruning methods (Carreira-Perpiñán & Idelbayev, 2018; Guo et al., 2016; Han et al., 2015) have become mainstream due to their relative low overheads and good scalability to large NNs. However, they still hinge on the expensive train-prune-retrain cycles to restore performance, and do not lead to efficient training. LTH (Frankle & Carbin, 2019) discovers sparse subnetworks at random initialization that can achieve similar or even better performance when trained in isolation. The winning tickets are found by the computationally intensive *iterative magnitude-based pruning (IMP)*, which involves tens of rounds of train-prune-retrain cycles. Rewinding was later found to be essential for stably identifying winning tickets in large networks (Frankle et al., 2019; 2020). (Morcos et al., 2019; Chen et al., 2020; 2021a;b) studied the transferability of winning tickets between datasets, tasks and architectures.

Zhou et al. (2019) made further interesting observations on winning tickets, discovering that preserving the original weight signs is essential to successfully training the tickets, and that the pruned weights have to be set to zero instead of being frozen at initial values. They concluded that "masking is training", i.e., the masking operation tends to move weights in the direction they would have moved during training. Hence, simply applying an appropriate mask to the random initialization leads to reasonably good performance. The authors found such "supermasks" with both heuristics and Bernoulli sampler-based optimization. Their finding was empirically concurred by (Ramanujan et al., 2020) and theoretically supported by (Malach et al., 2020; Pensia et al., 2020; Orseau et al., 2020). Similar results are also observed and proved for binary neural networks.

However, the original experiments in (Zhou et al., 2019) were restricted to NNs with a very small number of layers (e.g., Conv 4). Later work (Ramanujan et al., 2020) identified untrained subnetworks from Wide ResNet-50 as "mother NNs" on ImageNet that can match the performance of ResNet-34 (worse than ResNet-50). Yet that is very different from our (more ambitious) goal - pruning a subnetwork from the "mother NN", and having it match the original performance of the "mother NN" by only flipping-based training. Moreover, their sophisticated algorithm relies on a set of real-valued latent weights (called "scores" associated with each NN weight), making their training essentially as expensive as full SGD.

Another line of works has arisen to investigate pruning NNs directly at random initialization. Examples include SNIP (Lee et al., 2019), GraSP (Wang et al., 2020) and SynFlow (Tanaka et al.,

2020), all of which exploit the weight magnitudes as well as first- or higher-order information of NNs at the initial point. (Frankle et al., 2021) revisited these methods and observed that the naive magnitude-based pruning at initialization already plays as a strong competitor, with a side observation that delaying the pruning generally improves performance. All these algorithms train the pruned networks using the same standard SGD-type optimizers.

## 2.2 OPTIMIZATION BY SIGN FLIPPING

The train-by-flipping scheme considered in this work is closely related to the optimization of binary neural networks (BNNs), whose weights are only +1/-1. Optimization methods over BNNs can be categorized into two types – training with and without real-valued latent weights. The former category (Bengio et al., 2013; Courbariaux et al., 2016) maintains and updates a set of full-precision weights for back-propagation while forward passes only utilize binary weights. The downside of this method is the extra memory and resource overhead in training. The second category eliminates latent weights and only learns the decision rules for flipping or not, based on certain training statistics. For example, Bop (Helwegen et al., 2019) keeps a running average of historical gradients, and flips a weight's sign when its accumulated gradients surpasses a pre-defined threshold.

A more recent work (Ivan & Florian, 2020) studied the possibility of training NNs by only flipping the signs of the initialized weights, leaving the magnitudes unchanged. The authors proposed to update an extra set of latent parameters, whose signs determines the weight flipping decision throughout training. Their method struggled to scale up to large NNs, causing large performance drops. The potential reason behind this might be just as (Zhou et al., 2019) advocated, that zero would a particularly good value to set many weights to. In other words, weight sparsity may be a generally helpful inductive prior for well-trained NNs, which cannot be met by only flipping signs of weights. Besides, the training algorithm in (Ivan & Florian, 2020) also induces large overhead due to using full-precision latent weights and does not head towards efficient training. Moreover, neither (Ivan & Florian, 2020) nor any BNN optimization methods have considered the intersection of training-by-flipping and sparse NNs.

## 3 PEEK-A-BOO: SEARCH AND UNMASK DISGUISED SUBNETWORKS IN RANDOMLY INITIALIZED NEURAL NETWORKS

### 3.1 DISGUISED SUBNETWORKS VERSUS HIDDEN SUBNETWORKS

In this subsection, we formally define *hidden subnetworks* and *disguised subnetworks* in randomly initialized NNs, and show that the latter, as an extended class of subnetworks to the former, contain better subnetworks that can outperform those in the former.

Consider a neural network $f(x; \omega)$ parameterized by (vectorized) weights $\omega \in \mathbb{R}^n$ for input $x$. We denote the randomly initialized weights as $\omega^{(0)}$, and $\omega^{(t)}$ as the weights after $t$ steps of training. We denote a general transformation applied on the weights $\omega$ as $U(\omega)$ that takes any form.

**Definition 1.** *A hidden subnetwork in the dense neural network $f(x; \omega)$ is characterized by a binary mask $m \in \{0,1\}^n$ and denoted as $f(x; \omega \odot m)$, where $\odot$ represents the Hadamard product. A disguised subnetwork is defined as a transformed hidden subnetwork characterized by a binary mask $m \in \{0,1\}^n$ and a weight transformation $U(\cdot)$, denoted as $f(x; U(\omega \odot m))$*

**Remark** Here we omit the index for the number of training steps because we can take hidden subnetworks or disguised subnetworks at any time during training, while in this work our focus is on the subnetworks at random initializations, i.e., $f(x; \omega^{(0)} \odot m)$ and $f(x; U(\omega^{(0)} \odot m))$.

To identify the optimal disguised subnetworks at a certain sparsity level $S$, we need to solve the optimization problem below:

$$\underset{U \in \mathcal{U}, \; \|m\|_0 = S}{\text{minimize}} \mathcal{L}_D[f(x; U(\omega^{(0)} \odot m))], \tag{1}$$

where $\mathcal{U}$ is the collection of all possible weight transformations and $\mathcal{L}_D$ is the loss function on dataset $D$. It is clear that the optimal disguised network achieves a minimum that is equal to or smaller than that achieved by the optimal hidden network, because the hidden networks are the

special cases of disguised subnetworks if we constrain the weight transformation $U$ to the identity mapping $I$. And the optimal hidden subnetwork is the minimizer of the following problem:.

$$\underset{\|m\|_0=S}{\text{minimize}} \ \mathcal{L}_D[f(x; I(\omega^{(0)} \odot m))]. \tag{2}$$

In this paper, we argue that constraining the weight transformation $U$ to identity as in the hidden networks can significantly decrease the model capacity and thus degrade the performance, especially when $n$ is large, i.e., the model is large or $s$ is small, i.e., the model is highly sparse. In contrast, if we can directly solve (1), we can find disguised subnetworks with much better performance.

However, directly optimizing (1) is computationally intractable due to the high-dimensionality of the networks and the complexity of the collection of weight transformations. Therefore, we propose to approximately solve (1) by decoupling it into a two-phase process. In the first phase, we find a suitable mask $\hat{m}$ to sparsify a deep neural network at *its random initialization*. We call it the *searching* phase. In the second phase, we learn the optimal weight transformation to further lower the training loss. We call it the *unmasking* phase. The whole process is formally formulated as:

$$\hat{m} \in \underset{\|m\|_0=S}{\text{argmin}} R(f(x; \omega^{(0)} \odot m)) \tag{3}$$

$$\hat{U} \in \underset{U \in \mathcal{U}}{\text{argmin}} \mathcal{L}_D[f(x; U(\omega^{(0)} \odot \hat{m}))] \tag{4}$$

Here $R$ is a score function that measures the quality of the binary mask $m$ for the network $f$ with random initialization $\omega^{(0)}$ in some metric. Different selections of this score function will lead to different pruning methods. To efficiently solve this two-phase optimization in (3) and (4), we propose *Peek-a-Boo* (**PaB**) framework as detailed below.

## 3.2 Peek-a-Boo: efficient searching and unmasking

In the searching phase of the PaB framework, we sparsify a neural network at its random initialization using a training-free pruning method. In the unmasking phase, we learn to flip signs of the remaining nonzero weights as the weight transformation $U$ to lower the training loss.

**Searching phase: training-free pruning.** Our goal in the first step is to use pruning to find a sparse mask $\hat{m}$, with two requirements. Firstly, the sparse mask provides a useful prior of "good structures" in the network for the flipping-based training that follows. Secondly, the pruning itself needs to be light enough to not incur overhead for training efficiency. Due to the second concern, we give up the expensive IMP (Frankle & Carbin, 2019; Zhou et al., 2019) and focus on the pruning-at-initialization methods (Lee et al., 2019; Wang et al., 2020; Tanaka et al., 2020; Frankle et al., 2021).

We choose SynFlow (Tanaka et al., 2020) over SNIP (Lee et al., 2019) and GraSP (Wang et al., 2020), because the latter two depend on the first-order and/or second-order information of the neural network to calculate the weight scores. To accurately estimate those, one needs to run through training samples. In comparison, SynFlow is data-free and only uses an all-one input.

**Unmasking phase: flipping the non-zero weights.** In this step, we learn the optimal weight sign flipping $U_s(\omega) = \omega \odot s$ to lower the training loss. Here $s$ is a $\{+1, -1\}$ binary mask that we want to optimize. Allowing to flip the signs of the remaining nonzero weights, the unmasking phase significantly alleviates the limitation in model capacity. That is, one sparse NN of $k$ non-zero elements can be augmented to $2^k$ possible candidates, if sign flipping is enabled. However, it is computationally intractable to exhaustively search for the optimal vector. To approximately solve (4) with sign flipping transformation $U_s$, we leverage a prior art called Bop from BNN optimization (Helwegen et al., 2019). As formulated in Appendix A, Bop selects what weights and when to flip their signs by taking into account the exponential moving average of historical gradients. Different from the `edge-popup` algorithm used by Ramanujan et al. (2020) and a similar method in (Ivan & Florian, 2020), Bop gets rid of additional latent weights, hence incurring more training overhead and deviating from our end goal of more efficient training. Bop also avoids the use of *straight-through estimator (STE)* during backpropagation, which is known to unstabilize the training.

## 3.3 Flipping using cheap gradients: towards real computational efficiency

The training-by-flipping scheme only seeks to optimize the sign configuration, not the the real-valued weights (they are fixed as initialized). Hence, an educated guess is that this scheme can

tolerate "coarser" gradient information than standard training methods. This leaves decent room for us to further reduce the training cost by replacing precise gradients with approximated ones. One way to implement this is to adopt a low-precision gradient approximation called *Predictive Sign Gradient*, which was shown to save computation on hardware (Wang et al., 2019).

PSG predicts the signs of the precise gradients. using the signs of approximated gradients calculated from the *most significant bits (MSB)*. PSG only corrects its predictions when the approximated gradients are close to zero, e.g., implying a non-trivial chance of wrong predictions. Specifically, consider a one-layer neural network with input $x$, weight matrix $\omega$ and output $y = \omega x$. During back-propagation, we receive $g_y = \partial \mathcal{L} / \partial y$ for the proceeding layer, and calculate the weight gradient $g_\omega = \partial \mathcal{L} / \partial \omega = g_y x^T$ and the input gradient $g_x = \partial \mathcal{L} / \partial x = \omega^T g_y$. PSG replaces $(x, \omega, g_y)$ in the above calculations with their bit-quantized versions $(\tilde{x}, \tilde{\omega}, \tilde{g}_y)$, with bit-width $(b_x, b_\omega, b_g)$, and uses the resulting $\tilde{g}_\omega = \tilde{g}_y \tilde{x}^T$ and $\tilde{g}_x = \tilde{\omega}^T \tilde{g}_y$ as the approximated weight and input gradient. $\tilde{g}_x$ is further propagated to preceding layers to run the back-propagation.

In PaB, we use $\tilde{g}_\omega$ itself instead of its sign, either in flipping with latent weights or flipping with Bop. This is because without changing the weight magnitudes, we only have two actions — to flip or not to flip. Too much training variance will be introduced if we directly use $\text{sign}(\tilde{g}_\omega)$ to flip the weight signs, thus causing instability and difficulty in training. Moreover, we ignore elements in $\tilde{g}_\omega$ that are close to zero, rather than correct them with precise gradient calculation. This can reduce the training cost and also help to smooth the training process. Gradients to be ignored are selected using an adaptive thresholding technique in (Wang et al., 2019). This behavior is discussed in Section 4.4. The full PaB framework is summarized in Algorithm 1.

---

**Algorithm 1** PaB Efficient Training Framework

---

**Require:** Randomly initialized network $f(x; \omega^{(0)})$, dataset $D = \{x_i, y_i\}_{i=1}^{N}$, loss function $\mathcal{L}_D$, sparsity $S$, threshold $\theta$, total number of training steps $T$, PSG configuration $[b_x, b_\omega, b_g]$.
1:   $m \leftarrow \text{Prune}(f(x; \omega), S, D)$ with $\|m\|_0 = S$          $\triangleright$ Get the sparse mask by pruning
2:   $s^{(0)} \leftarrow (1, \ldots, 1); t = 0$             $\triangleright$ Initialize sign configuration and time step
3: **while** $t < T$ **do**
4:     $\tilde{g}^{(t)} \leftarrow \text{PSG}\left(\mathcal{L}_D, f(x; \omega^{(0)} \odot m \odot s^{(t)}), [b_x, b_\omega, b_g]\right)$    $\triangleright$ Approximate gradient using PSG
5:     $s^{(t+1)} \leftarrow \text{Flip}\left(s^{(t)}, \{\tilde{g}^{(i)}\}_{i=1}^{t}\right)$        $\triangleright$ Flip signs using historical gradient information
6:     $t = t + 1$
7: **end while**
8: **return** $f(x; \omega^{(0)} \odot m \odot s^{(T)})$

---

## 4 EXPERIMENTS

### 4.1 MAIN RESULTS ON CIFAR-10 AND CIFAR-100

**Comparison Methods**. We compare our proposed PaB method with the standard SGD training for the dense network (*Dense-SGD*) and the `edge-popup` algorithm (Ramanujan et al., 2020). For the non-flipping training for pruned networks, we choose standard *SGD* and *SignSGD* (Bernstein et al., 2018) as the baselines. For the proposed PaB methods, we compare: (1) *PaB-Latent*, which first prunes the network and trains the unpruned weights using the flipping with latent weights method in (Ivan & Florian, 2020) [1]; (2) *PaB-Latent-PSG*, the variant of PaB-Latent combined with PSG; (3) *PaB-Bop*, which trains the unpruned weights using Bop (Helwegen et al., 2019); and (4) *PaB-Bop-PSG*, the variant of PaB-Bop combined with PSG. We also introduce a variant of Dense-SGD as baseline called *Dense-SGD-Short*, where we cut down the training epochs of Dense-SGD to match the number of training BitOps of that in PaB-Bop-PSG.

**Experiment and evaluation settings:** We evaluate the performance of these PaB methods on the CIFAR-10 using large ResNet-18, ResNet-50 architectures, in contrast to previous papers on flipping based training (Ivan & Florian, 2020) and lottery tickets (Zhou et al., 2019) where their methods

---

[1]The latest manuscript of (Ivan & Florian, 2020) just reported a new result on ResNet-18. However, their baseline accuracy number (90.45%) is inferior to what is commonly used by the community (over 95% using the implementation in `https://github.com/kuangliu/pytorch-cifar`). Their flipping algorithm ends up ~3% accuracy drop on this baseline. Without access to their official codes for large models, we are not sure whether we could draw fair comparison with their results.

Table 1: Experiments on CIFAR-10 using ResNet-18 and ResNet-50. Models are initialized with Kaiming Normal distribution and pruned using SynFlow with pruning ratio 90% at initialization.

| Model | ResNet-18 | | | ResNet-50 | | |
|---|---|---|---|---|---|---|
| | Acc (%) | Size (MB) | BitOPs | Acc (%) | Size (MB) | BitOPs |
| Dense-SGD | 95.10 | 42.59 | 1.71Tr | 95.34 | 89.54 | 3.99Tr |
| Dense-SGD-Short | 92.52 | 42.59 | 0.30Tr | 86.19 | 89.54 | 0.55Tr |
| Edge-popup (Ramanujan et al., 2020) | 58.54 | 1.46 | 1.20Tr | 75.90 | 3.08 | 2.79Tr |
| SGD | 94.39 | 5.59 | 0.72Tr | 94.46 | 11.75 | 1.32Tr |
| SignSGD (Bernstein et al., 2018) | 93.15 | 5.59 | 0.72Tr | 93.61 | 11.75 | 1.32Tr |
| PaB-Latent (Ivan & Florian, 2020) | 91.82 | 1.46 | 0.72Tr | 89.80 | 3.08 | 1.32Tr |
| PaB-Latent-PSG | 91.84 | 1.46 | 0.30Tr | 90.25 | 3.08 | 0.55Tr |
| PaB-Bop (Helwegen et al., 2019) | 92.55 | 1.46 | 0.72Tr | 92.13 | 3.08 | 1.32Tr |
| PaB-Bop-PSG | 92.71 | 1.46 | 0.30Tr | 91.61 | 3.08 | 0.55Tr |

Table 2: Experiment results on CIFAR-10 and CIFAR-100 using WideResNet-28 as testbeds. All models are initialized with Kaiming Normal distribution and pruned using SynFlow with pruning ratio 70% at initialization. ∗ We perform a range of hyperparameter tuning on SignSGD and report the best accuracy we observe.

| Model | WideResNet-28 - CIFAR-10 | | | WideResNet-28 - CIFAR-100 | | |
|---|---|---|---|---|---|---|
| | Acc (%) | Size (MB) | BitOPs | Acc (%) | Size (MB) | BitOPs |
| Dense-SGD | 96.30 | 139.10 | 18.28Tr | 81.11 | 139.10 | 18.28Tr |
| Dense-SGD-Short | 92.29 | 139.10 | 2.03Tr | 77.04 | 139.10 | 4.20Tr |
| Edge-popup (Ramanujan et al., 2020) | 40.77 | 4.78 | 12.80Tr | 45.01 | 3.08 | 12.80Tr |
| SGD | 95.50 | 18.26 | 4.86Tr | 80.28 | 46.08 | 10.10Tr |
| SignSGD (Bernstein et al., 2018) | 94.18 | 18.26 | 4.86Tr | 77.82 | 46.08 | 10.10Tr |
| PaB-Latent (Ivan & Florian, 2020) | - | 4.78 | 4.86Tr | - | 4.78 | 10.10Tr |
| PaB-Latent-PSG | 89.29 | 4.78 | 2.03Tr | 68.90 | 4.78 | 4.20Tr |
| PaB-Bop (Helwegen et al., 2019) | 94.41 | 4.78 | 4.86Tr | 77.59 | 4.78 | 10.10Tr |
| PaB-Bop-PSG | 94.26 | 4.78 | 2.03Tr | 77.81 | 4.78 | 4.20Tr |

scale only up to 6-layer conv networks. In all experiments, we initialize networks using Kaiming Normal initialization (He et al., 2016), prune to 90% pruning ratio using SynFlow for 100 iterations (Tanaka et al., 2020) at random initialization, and train over 200 epochs. For Bop, we start with adaptivity ratio $\gamma = 10^{-3}$, which is decayed by 0.15 every 45 epochs, and use threshold $\tau = 10^{-6}$. For non-Bop methods and non-Bop weights, we use an initial learning rate of 0.1 and decay them by 0.1 at epoch 80 and 120. We use an initial learning rate of 0.01 for SignSGD as smaller learning rates are favored in (Bernstein et al., 2018). Results are presented in Table 1, including accuracies, model sizes and training BitOPs (in trillions) for each method and network. Huffman coding is used to encode the PaB models as described in Appendix D. We calculate training BitOPs following (Jin et al., 2020) instead of FLOPs because PSG involves low-precision computation.

**Main observations:** Several observations can be drawn from the results:

- As expected, the PaB framework results in impressive compression ratios ($> 30x$) on all models, thanks to the fact that flipping-based methods leave weight magnitudes unchanged, which can be restored easily using the same random seed.

- We see that PaB-Bop and its PSG variant are consistently better than PaB-Latent and the PSG variant. This shows the superiority of Bop-based methods over latent-weight methods (Ivan & Florian, 2020) in terms of the effectiveness when training sparse networks. We perceive this observation as an indication of the potential existence of even better sparse training methods than the Bop-based method that can exploit the sparse structure more effectively.

- The integration of PSG with PaB further reduces the training cost by significant margins, saving 82∼86% of the total BitOPs that is needed for training a dense counterpart.

- PaB-Bop-PSG performs comparatively well for ResNet-18/50, achieving $1.68\%$ and $2.33\%$ accuracy gaps respectively compared to SGD and even smaller gaps compared to SignSGD.

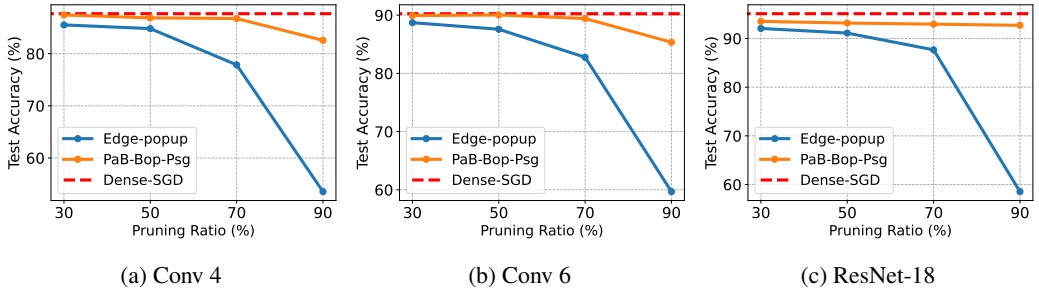

(a) Conv 4        (b) Conv 6        (c) ResNet-18

Figure 2: Comparing the performance of PaB-Bop-Psg and the edge-popup algorithm (Ramanujan et al., 2020) on Conv4, Conv6 and ResNet-18 on the CIFAR-10 dataset.

**Results on wider networks:** Besides scaling ResNet-18 up to ResNet-50 in depth, we also show the scalability of PaB in width by training WideResNet-28 with a 10x width multiplier, on both CIFAR-10 and CIFAR-100 datasets. As seen from the results shown in Table 2, PaB-Bop-PSG scales very well to WideResNet-28 on both CIFAR-10 and CIFAR-100. On CIFAR-10, PaB-Bop-PSG incurs a $1.24\%$ gap compared to SGD and even outperforms SignSGD. Similarly on CIFAR-100, PaB-Bop-PSG incurs a $2.47\%$ gap from SGD and performs similarly to SignSGD. The accuracy gaps of PaB on WideResNet-28 are even smaller than on ResNet-18/50, demonstrating that PaB scales well to wide networks, potentially due to better gradient flow in wider architectures. We omit PaB-Latent results here because we fail to train PaB-Latent to reasonable accuracies for WideResNet-28, which, in addition to the observed gaps to PaB-Bop on ResNet-18 and -50, shows the superiority of Bop-based methods over flipping with latent weights.

## 4.2 COMPARISON WITH STATE-OF-THE-ART EFFICIENT TRAINING METHODS

We compare PaB with two state-of-the-art efficient training methods, $E^2$-Train (Wang et al., 2019) and EB-Train (You et al., 2019). $E^2$-Train improves training efficiency by combining data dropping, model layer skipping, and gradient quantization, and EB-Train utilizes structured pruning and the early emergence of sparsity patterns. We train ResNet-18 on CIFAR-10 using all three methods with different efficiency-performance trade-offs. As shown in Figure 1, PaB achieves better accuracy-BitOps trade-off than $E^2$-Train and EB-Train, especially when high training efficiency is demanded (small BitOps). The x-axis is the average BitOps for training one sample for one step. We present the detailed protocol for this comparison in Appendix.

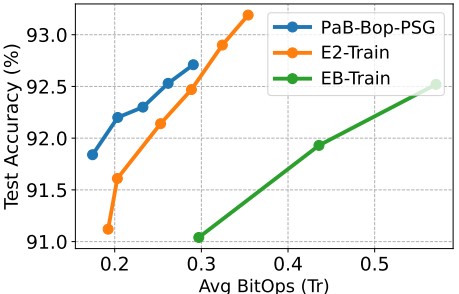

Figure 1: Comparison between PaB and state-of-the-art efficient training methods. PaB achieves better trade-off between training BitOps and accuracy.

## 4.3 COMPARISON WITH EDGE-POPUP ALGORITHM

We compare PaB with the `edge-popup` algorithm proposed in (Ramanujan et al., 2020) to identify the hidden subnetworks in deep neural networks at their random initializations. We adopt two small networks, Conv4 and Conv6[2] used in (Ramanujan et al., 2020) and a larger network ResNet-18 for comparison. All models are trained on CIFAR-10. We apply PaB and `edge-popup` to the testing networks with pruning ratios 30%, 50%, 70% and 90%. Results presented in Figure 2 show that PaB consistently finds better disguised subnetworks than the hidden networks found by `edge-popup` algorithm, especially when the pruning ratios are high.

We further implement PaB on the ResNet-50 for **ImageNet** and compare with `edge-popup` (Ramanujan et al., 2020). For both methods we train the ResNet-50 network for 100 epochs, following the same settings as in (Ramanujan et al., 2020). With 30% and 50% sparsity ratio (the percentage of pruned parameters), PaB-PSG can achieve 63.58% and 63.25% accuracy, in contrast to the 39.5% and 58.4% accuracy achieved by `edge-popup`.

---

[2]We follow the official implementation of the edge-popup algorithm but make slight modifications to the Conv4 and Conv6 networks by adding batch normalization layers to stabilize the training.

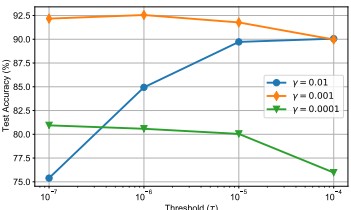

Figure 3: PaB-Bop accuracy on CIFAR-10 using different hyperparameter combinations of $\gamma, \tau$ with ResNet-18. Networks are pruned using SynFlow with 90% pruning ratio.

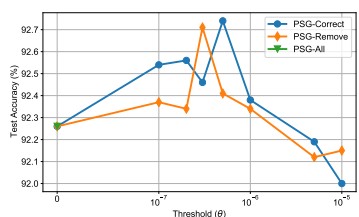

Figure 4: PaB-Bop-PSG accuracy on CIFAR-10 using different PSG behaviors and thresholds. Networks are pruned using SynFlow with 90% pruning ratio.

### 4.4 Ablation studies on PaB

Due to space limit, we only present the ablation study on two important sets of hyperparameters in PaB, namely the hyperparameters for BOP, and those that control the small gradient behavior in PSG. We leave more ablation studies on the pruning ratio, the epoch when pruning occurs, and the random distribution for initialization to the Appendix. We use ResNet-18 and CIFAR-10 by default.

**BOP hyperparameters:** Different from PaB-Latent, which in nature still trains latent weights using SGD, PaB-Bop consists of two parts: the accumulation of historical gradients and a thresholding mechanism to select significant updates in the correct directions, controlled by adaptivity ratio $\gamma$ and threshold $\tau$, respectively, as formulated in (6). Therefore, we conduct an ablation study on $\gamma$ and $\tau$ and present the results in Figure 3. We prune ResNet-18 networks with pruning ratio 90% using SynFlow and then train them on CIFAR-10 using PaB-Bop with different combinations of $\gamma$ and $\tau$. When $\gamma$ is small, the latest gradient makes too small contributions in the accumulation to cause enough updates. In contrast, when $\gamma = 0.01$, Bop needs a larger threshold $\tau$ to reduce the training variance. The best combination is $(\gamma, \tau) = (10^{-3}, 10^{-6})$.

**Dealing with small gradients in PSG:** In the original PSG (Wang et al., 2019), when PSG obtains small gradient approximations (judged by a threshold $\theta$), PSG corrects them with precise gradients using full-precision calculation. Here we study three different PSG behaviors in the framework of PaB: (1) *PSG-All*, where we take whatever gradient approximated; (2) *PSG-Correct*, the default behavior in (Wang et al., 2019), which corrects small approximated gradients with precise ones; and (3) *PSG-Remove*, which removes small approximated gradients. The accuracy results of applying PSG with these three different behaviors to ResNet-18 pruned by SynFlow are shown in Figure 4. Note that PSG-Correct/Remove reduces to PSG-All with zero threshold. The results show that PSG-Correct and PSG-Remove perform similarly at their best thresholds. Since PSG-Correct requires extra computation of precise small gradients, we prioritize efficiency and use the PSG-Remove behavior in all PSG experiments.

## 5 Conclusion, Future Works, and Discussion of Societal Impacts

We study Peek-a-Boo (PaB) as a new minimalist approach to training dense networks at their random initializations without altering weight magnitudes. Inspired by the recent exploration of sparse NNs, PaB achieves high training efficiency and provides a natural lossless compression after training. While this work remains as a pilot study, we show it can already scale up to larger networks and datasets (e.g, up to ResNet-50 and ImageNet) with competitive trade-off between accuracy and training efficiency. We hope that PaB will evoke more theoretical reflections on training. One of our immediate future work will explore alternatives to the sign flipping operation as the weight transformation in the unmasking phase. For example, we can extend sign flipping with learnable scaling, or replace it with quantization. In another direction, while this paper mainly uses unstructured sparsity following the convention of pruning at initialization, we can explore how structured sparsity (You et al., 2019) may fit in PaB for practical hardware acceleration.

### Acknowledgment

Z.W. is supported by a US Army Research Office Young Investigator Award (W911NF2010240) and an NSF SCALE MoDL project (2133861).

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

## A  BOP FORMULATION

Bop (Helwegen et al., 2019) selects what weights and when to flip their signs by taking into account past gradient information. In particular, Bop accumulates the exponential moving average of historical gradients:

$$e^{(t+1)} = (1 - \gamma)e^{(t)} + \gamma\nabla_\omega\mathcal{L}_D[f(\omega^{(t)})], \tag{5}$$

where $\gamma$ is called the adaptivity ratio. Bop flips weight signs when the moving average surpasses a certain threshold $\tau$ and aligns with the weights in direction:

$$w^{(t+1)} = \begin{cases} -w^{(t)} & \text{if } |e^{(t)}| \geq \tau \text{ and } e^{(t)}w^{(t)} > 0, \\ w^{(t)} & \text{otherwise.} \end{cases} \tag{6}$$

## B  PROTOCOL OF COMPARING PAB WITH E$^2$-TRAIN AND EB-TRAIN

EB-train (You et al., 2019) is easier to control the computational costs and calculate its BitOps. EB-Train consists of three stages: ($i$) the searching stage where a dense model is trained with sparsity regularization applied on the scaling factors of the batch normalization layers; ($ii$) prune the filters when the convergence of the pruning mask is detected; ($iii$) continue training the pruned models. In the first stage, EB-Train has the same computational cost as a dense model, but only for a short period of training. After the pruning is performed in the second stage, we can easily calculate the reduced BitOps based on the layerwise pruning ratios induced by pruning. Such reduced computational cost will be sustained throughout the third stage. In the experiment shown in Figure 1, we change the pruning ratio in EB-Train to control the BitOps.

E$^2$-Train Wang et al. (2019) is a more tricky baseline because it consists of three components - *stochastic mini-batch dropping* (SMD), *selective layer update* (SLU) and *predictive sign gradient* (PSG). We follow the setting of SMD ratio, i.e., the ratio of randomly dropped mini-batches during training and fix it at 50%. The BitOps reduction brought by PSG is decided by the bit-width of the most significant bits. We use the same 8-8-16 bit widths for inputs/activations, weights and gradients respectively as in (Wang et al., 2019) and use full-precision computations for the forward passes. We then change the computation budget in SLU to control the BitOps in Figure 1.

Although the computation reduction of PaB is fixed given the sparsity ratio and the bit-width configuration in PSG, PaB is orthogonal to stochastic mini-batch dropping and can be combined with it. Hence, we change the ratio of dropping mini-batches to control the BitOps of PaB-Bop-PSG in Figure 1.

## C  ADDITIONAL ABLATION STUDIES

**Pruning time point**. Although pruning at initialization has shown success at training sparse networks, (Frankle et al., 2021) identify that pruning at different time points after initialization could further improve the performance of these networks. Using a fixed pruning ratio of 90%, we train ResNet-18 on CIFAR-10 and observe the effects on the test accuracy of delaying pruning to different training points from epoch 0/200 to epoch 60/200. Figure 5a shows the performance of our non-flipping baselines and PaB methods as the pruning epoch varies. Different from the observations in (Frankle et al., 2021), while slight increases in accuracy can be observed with delayed pruning, the effect is quite small at the scale of this experiment. The reason might be that we fix the total number of epochs whenever we perform pruning, while (Frankle et al., 2021) goes through the whole training process after pruning. With efficient training in mind, we maintain pruning at initialization in subsequent experiments since delaying pruning after initialization would require expensive training of dense networks before pruning.

**Pruning ratios**. We also evaluate the performance of our PaB methods against our baselines at different levels of sparsity. Figure 5b shows the test accuracies from training ResNet-18 on CIFAR-10 at varying pruning ratios from 0% to 90%. Not surprisingly we see a decreasing tendency in test accuracy when we gradually increase the pruning ratios for all methods, especially going from 80% to 90%. However, PaB-Bop and its PSG variant are more robust to high pruning ratios, while

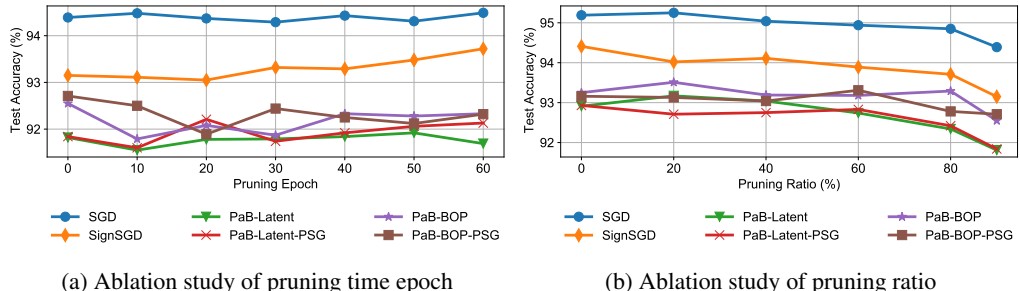

(a) Ablation study of pruning time epoch      (b) Ablation study of pruning ratio

Figure 5: Accuracy of baseline and PaB methods for ResNet-18 and CIFAR-10 with different pruning time epochs (0-60 epoch) and pruning ratios (0-90%).

PaB-Latent and PaB-Latent-PSG have obvious degradation at 90% pruning ratio. Beyond 90%, the test accuracy starts to drop noticeably, and the resulting sparse networks becoming untrainable for all training methods due to layer collapse. By default hereinafter, we fix the pruning ratio to 90%.

**Ablation on initialization**. In previous experiments, we use Kaiming Normal distribution for random initialization. We also try Kaiming Uniform and Xavier Normal (Glorot & Bengio, 2010) distributions as alternatives. As shown in Table 3, there is generally little variation in performance across initialization methods, but we are able to achieve the best results across all PaB methods from PaB-Bop-PSG initialized using the Kaiming Normal distribution. We can also see that all PaB methods consistently perform better under Kaiming Normal initialization than under Xavier Normal initialization.

Table 3: Accuracy on CIFAR-10 with different initialization for ResNet-18. All networks are pruned using SynFlow at initialization with 90% pruning ratio.

| Method | Kaiming Normal | Kaiming Uniform | Xavier Normal |
|---|---|---|---|
| SGD | 94.39% | 94.63% | 94.31% |
| SignSGD | 93.15% | 93.41% | 93.39% |
| PaB-Latent | 91.82% | 92.12% | 91.75% |
| PaB-Latent-PSG | 91.84% | 92.08% | 91.47% |
| PaB-Bop | 92.55% | 92.62% | 91.91% |
| PaB-Bop-PSG | 92.71% | 92.53% | 92.21% |

**Iterative pruning and sign-flipping training**. Inspired by the iterative scheme of IMP, which is observed to be essential for finding winning tickets, we extend SynFlow from one-shot pruning to multiple-shot over a period of the training process. Note that one shot of SynFlow can contain many iterations of pruning as described in Tanaka et al. (2020). Here by "multi-shot" we mean to apply SynFlow several times throughout the training process to gradually increase the sparsity level. Specifically, instead of applying SynFlow to the networks once at initialization, we run SynFlow for 2~5 times, evenly spread over the first 50 epochs, so that the sparsity ratio linearly increases to a certain level (90%). Between two runs of pruning or after the last run of pruning, we perform the flipping-based training using Bop.

This iterative multi-shot version of PaB can be seen as alternating between the optimization processes of the searching and unmasking phase to discover the disguised networks. Compared to the one-shot variant, multi-shot PaB is closer to a joint optimization scheme on the objective in (1). The results are shown in Table 4. We observe that such an alternative scheme provides little accuracy improvement, while apparently losing more of training efficiency. We do not intend to say that the joint optimization is not good: to properly solve the joint optimization problem in (1) with iterative algorithms is interesting and potentially desirable. Our argument is that, with our goal to do "efficient" finding of disguised subnetworks, our current two-step relaxation is not only the most efficient; but also highly competitive in accuracy with no sacrifice.

**Changing the bit-widths in PSG**. In PSG, the bit-width configuration $(b_x, b_\omega, b_g)$ for layer inputs, layer weights and output gradients controls the trade-off between training efficiency and the precision of approximated gradient, and thus the model performance. We apply PaB-Bop-PSG with varying bit-configurations and show the results in Table 5. We observe that the bit-width of output gradient $b_g$ has a strong influence on the final performance, which is consistent with previous low-precision efficient training works (Wang et al., 2019). We choose $(b_x, b_\omega, b_g) = (8, 8, 16)$ as it is a combination of good performance and training efficiency.

Table 4: Results of the iterative multi-shot variant of PaB. We run SynFlow for 2∼5 times, evenly spread over the first 50 epochs, so that the sparsity ratio linearly increases to a certain level (90%). Between two runs of pruning or after the last run of pruning, we perform Bop training.

| Runs of SynFlow | 2 | 3 | 4 | 5 |
|---|---|---|---|---|
| SGD | 94.79% | 94.76% | 94.48% | 94.68% |
| SignSGD | 93.20% | 93.57% | 93.51% | 93.77% |
| PnF-Latent | 91.98% | 92.05% | 92.00% | 92.29% |
| PnF-Latent-PSG | 91.93% | 91.94% | 92.06% | 91.89% |
| PnF-Bop | 92.25% | 92.18% | 92.15% | 92.48% |
| PnF-Bop-PSG | 92.28% | 91.89% | 92.25% | 92.32% |

Table 5: Accuracy and training cost in BitOPs on CIFAR-10 with varying bit-width configurations for ResNet-18. All networks are pruned using SynFlow at initialization with 90% pruning ratio.

| Configuration | (4,4,4) | (4,4,8) | (8,8,8) | (8,8,16) | (16,16,16) | (16,16,32) |
|---|---|---|---|---|---|---|
| Accuracy (%) | 32.38 | 89.68 | 89.02 | 92.71 | 92.24 | 92.29 |
| BitOPs | 0.25Tr | 0.25Tr | 0.27Tr | 0.30Tr | 0.36Tr | 0.48Tr |

## D  HUFFMAN CODING FOR ENCODING MASK AND SIGN ENCODING

PaB methods, without changing the magnitudes of the random initialization, only induces two outputs that need to be stored - ($i$) the sparse mask generated by the pruning methods (SynFlow in this work); and ($ii$) the sign flipping actions for all weights. These two outputs can be merged into one format, where the pruned weights are represented by $0$, sustaining the sign by $+1$ and flipping the sign by $-1$. Considering that the pruned weights account for the majority of all weights (90% in our CIFAR-10 experiments and 70% in our CIFAR-100 experiments), we can adopt Huffman coding to encode $0$ using 1-bit representation, and the sign flipping actions with 2-bit representation, as illustrated in Figure 6.

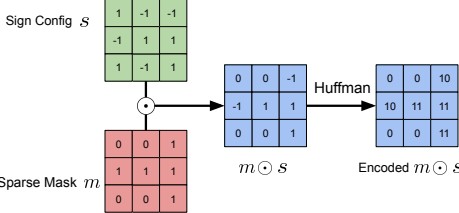

Figure 6: Huffman encoding for PaB model compression.

