# OpenReview forum: "Peek-a-Boo: What (More) is Disguised in a Randomly Weighted Neural Network, and How to Find It Efficiently"
_ICLR.cc/2022/Conference — ICLR 2022 Poster_

### Official Review · Reviewer_rMZy · 2021-10-27

**Correctness:** 3
**Technical Novelty And Significance:** 3
**Empirical Novelty And Significance:** 2
**Recommendation:** 5
**Confidence:** 3

**Main Review:**

This paper presents an algorithm named peek-a-boo (PaB) to optimize network pruning (at initialization) and optimization (limited within flipping the sign of weights). This setting has not been studied by prior works. A two-step algorithm was designed -- pruning first, optimization second. Experiments show competitive performance compared to prior methods with similar optimization complexity.

Though the setting is new and interesting, I shall say that its value to the community has not been perfectly revealed based on the existing experiments. In the studied cases, the neural networks and datasets are mostly small. The value of this work is supposed to be accelerating large-scale network training, but I am not sure that it scales up well.

Second, it seems that the technical contribution of the proposed algorithm is limited. When I see the objective function (1), I was expecting that pruning and optimization are jointly performed, however, according to Algorithm 1, they were performed separately, and the pruning part follows an existing algorithm. Separate optimization brings the issue that the best pruned architecture may not fit very well in the unmasking setting. Therefore, I am wondering if there is an iterative solution, that pruning is divided into several steps (e.g. one decides to keep 50% weights, so he/she can prune 10% weights every time for 5 rounds), and after each step, the flipping operation is performed. This can align both parts and hopefully improve the final performance. Of course, this may increase the training BitOps, but this shall be a tradeoff -- anyway, showing whether this strategy helps is not a bad trial.

Third, the experiments are performed on CIFAR datasets. Since ImageNet results are missing, I am not sure if the approach can actually scale up. In particular, according to Tables 1 and 2, PaB does not show a significant advantage over SGD, which further limits the application of the proposed approach. I think ImageNet experiments might be a good add-on to this paper.

**Summary Of The Paper:**

This paper presents an algorithm named peek-a-boo (PaB) to optimize network pruning (at initialization) and optimization (limited within flipping the sign of weights). This setting has not been studied by prior works. A two-step algorithm was designed -- pruning first, optimization second. Experiments show competitive performance compared to prior methods with similar optimization complexity.

**Summary Of The Review:**

1. The setting is new, but the practical value to the community seems unclear.
2. The algorithm is straightforward, joint optimization is not considered.
3. Experiments are promising but not exciting, ImageNet results are absent.

Overall, a bit below the acceptance threshold.

---

> ### Author Response · Authors · 2021-11-18
> **Response to Reviewer rMZy**
>
> Dear reviewer rMZy,
>
> Thank you for your constructive comments.
>
> ## Scaling up the PaB framework
>
> As it seems to be your major concern on our proposed PaB framework, we quickly implemented PaB on the ResNet-50 for ImageNet and conducted experiments to compare with edge-popup (Ramanujan et al., 2019). For both methods we train the ResNet-50 network for 100 epochs, following the same settings as in (Ramanujan et al., 2019). With 30% and 50% sparsity ratio (the percentage of parameters that are pruned), PaB-PSG can achieve 63.58% and 63.25% accuracy, in contrast to the 39.5% and 58.4% accuracy achieved by edge-popup. We see clear improvements of PaB over edge-popup. We believe this helps to show the scalability of PaB.
>
>
> ## Joint or separate optimization for PaB?
>
> Firstly, we would like to humbly point out that the decoupling of the pruning and optimization phases is exactly the point of PaB, which results in the proposed separated algorithm. Besides its superior performance over the baseline methods (e.g., edge-popup), the efficient implementation of the algorithm to solve the joint optimization is also a major goal of this work. The decoupling enables high efficiency in PaB because (1) it makes both pruning and sign-flipping phases easier. Therefore, we can use a much computationally lighter algorithm (e.g., SynFlow) for pruning other than expensive methods like edge-popup; (2) the sequential process of pruning first and then flipping the signs allows us to utilize the benefits of sparse neural networks starting from step 1 in the second step.
>
> Secondly, we also quickly implemented the alternative optimization scheme as suggested and ran some experiments on the CIFAR-10 dataset to verify its effectiveness. Specifically, instead of applying SynFlow to the networks once at initialization, we run SynFlow for multiple times, evenly spread over the first 50 epochs, so that the sparsity ratio linearly increases to a certain level (90\%). Between two runs of pruning, we perform the flipping-based training using Bop. We evaluated the performance of this alternative version of PaB with 2-5 pruning times. The results are shown below. Note that pruning time being one is the one-shot PaB used in the paper. We observed that such an alternative scheme provides little accuracy improvement, while apparently losing more of training efficiency.
>
> | Pruning Time |      2 |      3 |      4 |      5 |
> |:------------:|-------:|-------:|-------:|-------:|
> |   Accuracy   | 92.25% | 92.18% | 92.15% | 92.48% |
>
> We do not intend to say joint optimization is not good:  to properly solve the joint optimization problem in eqn (1) with iterative algorithms is interesting and potentially desirable. Our argument is that, with our goal to do “efficient” finding of disguised subnetworks, our current two-step relaxation is not only the most efficient; but also highly competitive in accuracy with no sacrifice.
>
>
>
>
> ## Comparison to SGD
>
> As PaB is intended as a generalized framework that extends the type of optimization methods that do not update the weight magnitudes, we believe it would be more fair to compare PaB with edge-popup (Ramanujan et al., 2019). In contrast, SGD updates the magnitudes of the remaining weights at the very beginning of the training process.

---

> ### Author Response · Authors · 2021-11-28
> **Further comments and discussions will be appreciated!**
>
> Dear Reviewer rMZy,
>
> Thank you for your valuable time to review this work and the constructive comments. We posted our response to your comments a while ago, and are wondering if you could kindly share some of your thoughts so we can keep the discussion rolling to clarify your further questions if there are any.
>
> In our response,
>
> 1. We provided the empirical results of applying PaB to the ImageNet dataset to show its scalability. We observed clear superiority over edge-popup.
>
> 2. As you suggested, we quickly implemented an iterative optimization process that alternates between pruning and Bop training, with the pruning ratio gradually increasing. We observed no clear improvement over the separate two-step optimization process. While we do not intend to say that the joint optimization is not good, we humbly argue that the two-step relaxation is already effective enough and efficient at the same time.
>
> We would appreciate it if you look through our response and see if you have any further questions, for which we will be very grateful and glad to address!
>
> Best,
>
> Authors

---

### Official Review · Reviewer_jduC · 2021-11-02

**Correctness:** 4
**Technical Novelty And Significance:** 4
**Empirical Novelty And Significance:** 4
**Recommendation:** 8
**Confidence:** 4

**Main Review:**

**Strength:**
- This paper is very well written and a pleasure to read.

- The paper tackles an important question, which is to rethink the optimization strategies of sparse neural networks, especially when using the idea of masking as training. The observation that when we are only looking to flip signs of random weights during training, we can tolerate much coarser gradients is insightful. The results are significant as it reduces training cost compared to fully trained networks while maintaining competitive performance.

- The method is compared to an appropriate set of baselines over a reasonable set of model architectures.

**Weakness:**
- How exactly is the compression ratio calculated? The authors mention that this is due to the fact that the random initialization can be stored using a single seed. While this may be true in some cases, I would be hesitant to rely on this attribute as the seed may not result in the same parameters in different machines. It would be more reasonable to assume that the weight values must be stored for a more conservative estimate.

- Most of the work in supermasks consider the "signed constant" variant, where the weights are converted to a single signed value for each layer. This parameterization has often shown improved performance over the parameterization using actual weight values, and leads to significant reductions in model complexity and size. How would this variant work under the PaB framework?

**Summary Of The Paper:**

This paper proposes the idea of a "disguised subnetwork", which are hidden random subnetworks that can be transformed into a well-performing subnetwork. The paper introduces PaB as a way to uncover these subnetworks, by first searching for a mask over the random weights using pruning-at-initialization techniques, and then learning a transformation on the subnetwork. The paper further shows that this PaB process can be efficiently implemented, offering significant advantage over prior work.

**Summary Of The Review:**

The method is novel and achieves better efficiency-performance tradeoff than the various baselines. It tackles the question of rethinking optimization strategies for sparse NN training, which I think is an important research direction. I think the contributions of this paper are significant and would support its acceptance.

---

> ### Author Response · Authors · 2021-11-18
> **Response to Reviewer jduC**
>
> Dear Reviewer jduC,
>
> Thank you for your positive feedback.
>
> ## How exactly is the compression ratio calculated?
> Trying to relieve your concern about using the random seed for the weight representation, we can assure you that we have successfully implemented and verified it in our experiments. We are also aware of works that adopt the similar methodology in federated systems which contain various types of models [1]. Besides this, we believe that it is also possible to do some post verification using hashing techniques.
>
> If we are really assuming that full weights values have to be stored, then it has the same compression ratio as the normal pruning methods, which is approximately equal to the sparsity ratio. That will somehow discount the contribution of immediate compression after training, but will not hurt our benefit of high training efficiency as well as accuracy.
>
>
> ## How does the “signed constant" variant work in the framework of PaB
>
> Thank you for your suggestions. Firstly we note that applying SynFlow to the signed constant variant will result in trivial pruning mask because all parameters in one layer will have exactly the same score. However, replacing SynFlow with other pruning-at-initialization methods (e.g., SNIP and GraSP) can make the framework of PaB co-work the signed constant initialization. We ran the experiments using Kaiming Constant initialization for all layers (as used in Ramanujan et al., 2020) and did see comparable performance as Kaiming Normal and Kaiming Uniform initialization.
>
> ## Reference
>
> [1] Li, Ang, et al. "FedMask: Joint Computation and Communication-Efficient Personalized Federated Learning via Heterogeneous Masking." Proceedings of the 19th ACM Conference on Embedded Networked Sensor Systems. 2021.

---

### Official Review · Reviewer_7V6H · 2021-11-02

**Correctness:** 4
**Technical Novelty And Significance:** 2
**Empirical Novelty And Significance:** 2
**Recommendation:** 6
**Confidence:** 4

**Main Review:**

I think the paper is extremely well-written. There are no typos, the language and the presentation are very strong, and the flow is very fluent. Moreover, the literature is summarised very precisely and it includes most of the relevant papers to discuss, to the best of my knowledge. The algorithm is described well, and the contribution is discussed in a fair way without unrealistic claims.

My major concerns are as follows:

1 - **Contribution.** Problem (2) formally describes the “optimal hidden subnetwork” problem, and it immediately follows from the definition of problem (1) that what the authors suggest will be an ‘improvement’ over the training data. This is not unimaginable, that is, it is already known that any transformation of weights we optimise over the training data will be an improvement.

2 - **Novelty.** As discussed above, rather than the definition of disguised subnetworks, what matters most here is, (i) the classes of “U” (transformation) that can be handled in this work, and (ii) novelty of the solution algorithm the work proposes. Regarding these two points:
-  (i) the class of U is constrained to sign flips, which is already well-studied and being applied in similar settings;
- (ii) the solution algorithm splits the problem into two sub-problems. The first sub-problem solves for “m” (the masking variable) by using existing training data-independent methods (cf. SynFlow). After fixing “m”, the algorithm simply proceeds with the Bop method from the BNN optimisation literature. My concern here is that, since in the loss function the order of these variables is loss [ U( .. m ) ], first solving for "m" and then for "U" is the most trivial way, and while doing this using all of the well-known results is not bringing much novelty.

3 - **Generality.** The paper is defining a very general problem of optimal disguised subnetworks and mentioning this in the title/abstract/ and throughout the paper. However, in my view, this work is more like a case-study where heuristic methods are being used iteratively. Namely, the contribution is actually: “taking a recent paper that finds a masking variable, and just flipping the signs w.r.t. the training loss”. This is still an interesting result, but I also would like to highlight the distinction from my perspective. Finally, I would like to highlight that, although the optimization formulation of "disguised subnetworks" finds U (transformation) and m (masking) jointly, the relaxation proposed (PaB) first splits the problem so that the solution algorithm first finds a sub-network (i.e., m is found) and then changes the weights. So this algorithm does not find a disguised subnetwork (as claimed in the introduction), rather finds a subnetwork and then re-assigns the weights, which is *a* disguised subnetwork, but not a new architecture. I believe the problem definition of disguised networks is very general but PaB first finds the hidden architecture so this is not a new way of finding subnetworks.

4 - **Numerical Experiments.** This is a question for the author(s) rather than a direct concern: some of the algorithms in the numerical experiments, where PaB is being compared with, are training-independent algorithms. The fact that they are training-independent (or sometimes called “without training methods”) is the main focus in the literature around them. Malach et al. (2020), on the work(Ramanujan et al., 2019) comment: “within a sufficiently overparameterized neural network with random weights (e.g. at initialization), there exists a subnetwork that achieves competitive accuracy”. Is this mentioned in the paper? Otherwise, the fact that here there is an optimisation procedure after fixing the sparsity structure based on the training set already changes the setting of the problem here and introduces a bias in comparison?

However, I still think the idea of combining these existing methods to find a solution for the optimal disguised subnetwork problem (and hopefully a good one), can pave the way for a new research direction. Here the key standpoint is, rather than the generality or the novelty of solutions, to demonstrate that instead of solving a dense network and “transforming the weights”, or to mask for a sparse network, one can combine these methods to have a sparse network that also performs better than the alternatives. The numerical experiments are conducted and reported carefully, and the results look promising. Hence, despite the lack of theoretical contributions, I would like to give my decision as "marginally below the acceptance threshold".

Minor comments:
- Page 2: “Even worse is” maybe better to re-write?
- Page 3: “Unfortunately, the original experiments in (Zhou et al., 2019)” maybe change ‘unfortunately’?
- Page 5: In the first sentence, “s” is not capital but previously it was. The same happens again in the “Unmasking phase” part of S3.2.
- “Computationally infeasible” -> “computationally intractable”
- “ = argmin “ -> “ \in argmin” (or prove that argmin set is a singleton)
- Page 5: “For example, one sparse NN of k non-zero elements  […]” is this called an example because k is named? Maybe it is better to connect with before “, that is, if a sparse NN …”?
- Page 5: Optimisation over 2^k many vectors is called “NP-hard”. I am not sure but this terminology may be misleading. The optimisation problem itself can be tractable but there the input is just intractably large in “k”.  I am not very sure about whether it is common to say the problem is NP-hard. For example, with that logic, max{a_1, a_2, … , a_n }, that is found in linear time, is NP-hard in log(n)?
- Page 7: “and the PSG variant,.” There is an additional comma (,) before the full stop (.)


**Summary Of The Paper:**

This paper extends the definition of the hidden subnetworks in randomly initiated neural networks. The new notion of subnetworks, namely the disguised subnetworks, apply a transformation on the hidden subnetwork weights to obtain final weights (a posteriori finding the hidden subnetwork).

Mathematically speaking, the main decision variable of the underlying optimisation problem of finding “hidden subnetworks” is the so-called masking variable that is a binary vector, the variable that decides which components of the randomly initiated weights will be zero, constrained to a desired level of sparsity. On the other hand, the underlying optimisation problem of “disguised subnetworks” has an additional variable U that applies a transformation on the set of weights after being ‘sparsified’. The selection of U = I, for I being the identity transformation, recovers the problem of hidden subnetworks, showing that the latter is a generalisation.

The author(s) present a heuristic algorithm that solves the forenamed optimisation problem. The idea is to first find a solution for the masking variable where (i) U = I is taken, (ii) the objective function is independent of a training set. Afterward, the algorithm proceeds by solving the problem for U given the solution of the previous step, where the space of U is restricted to the class of transformations where only sign-flips are allowed. If we think of the above solution process as a two-phase problem, the author(s) use the literature on sparse neural networks for the first phase and use the literature on binary neural networks for the sign flipping phase.

**Summary Of The Review:**

**Pros**
- The paper has a very clear overview of the relevant research, and in general, the paper is written very well
- The numerical experiments are very thorough
- The idea of generalizing the notion of hidden subnetworks is relevant
**Cons**
- The contribution is limited -- the definition of disguised networks is straightforward and the solution method comprises a subsequent application of existing algorithms (not parallelized or nested, but applied sequentially)
- The optimal disguised networks problem is restricted iteratively so that the end result is not very interesting anymore

---

> ### Author Response · Authors · 2021-11-18
> **Response to Reviewer 7V6H (Part 1)**
>
> Dear reviewer 7V6H,
>
> Thank you for your detailed and constructive comments. We sincerely appreciate it and believe they will help to significantly improve the quality of our work. We thank you for your appreciation in the quality and clarity of our write-up. We try to address your concerns in our work below:
>
>
> ## Contributions, Novelty and Generality
>
> After reading your comments in the contribution, novelty and generality, we think two of your major concerns are:
>
> 1. The proposed framework of “disguised networks” is “imaginable”.
> 2. Although the concept of “disguised networks” is general as a joint optimization framework, the implemented algorithm (PaB) is trivial and suboptimal because it solves the two subproblems separately, making this work look heuristic.
>
> To address your concerns, we would like to first reiterate that, while we are interested in the theoretical understanding of the phenomenon, this work is originally intended to empirically improve the “without training methods” such as edge-popup in (Ramanujan et al., 2019) and find alternative algorithms that can optimize sparse NNs **efficiently*. We attribute the limited performance of edge-popup to reasons. Firstly, the optimization problem itself is too constrained by only allowing optimizing the sparse mask. Secondly, the implementation of the edge-popup algorithm is not good because it uses simple binarization operations and STE for backpropagation.
>
> The contributions of this work are closely related to the two limitations. To address the limitation in the optimization formulation, we propose the concept of “disguised networks”, which mathematically guarantees a better optimal solution than only optimizing the sparse mask. It “enlarges” the capability of the problem. Further, the two-step relaxation and the exploitation of Bop optimizer is designed for the second limitation in the optimization process. This relaxation, as you suggested, will “sacrifice” the capability of overall optimization, as a one-step truncation of the underlying iterative optimization problem. However, it brings the benefits of efficiency in the training process.
>
> We are well aware that the two-step relaxation of the joint optimization might lead to a suboptimal solution. To quickly verify if we can obtain improvements in performance by solving the optimization more jointly, we implement an iterative optimization process where we alternate between pruning and flipping-based training and gradually increase the sparsity. More specifically, we run SynFlow for 2-5 times, evenly spread over the first 50 epochs, so that the sparsity ratio linearly increases to a certain level (90\%). The results are shown below. Note that pruning time being one is the one-shot PaB used in the paper. We can see that such an alternative scheme provides little accuracy improvement.
>
> | Pruning Time |      2 |      3 |      4 |      5 |
> |:------------:|-------:|-------:|-------:|-------:|
> |   Accuracy   | 92.25% | 92.18% | 92.15% | 92.48% |
>
> We would also like to emphasize that the efficient implementation of the proposed algorithm is one of our major goals. The two-step relaxation enables high efficiency because we create and fix a sparse structure in the first step (pruning) and the second step can utilize the benefits of sparsity from the very beginning. The superior efficiency of PaB is clearly shown in Table 1 and 2, where Pab uses only 20~30% BitOps compared to edge-popup. It is also noteworthy that the above extension of iterative optimization will make PaB less efficient because it has lower sparsity at early training while providing little improvement in performance.
>
> Note that we still think an ideal algorithm that properly solves the joint optimization problem in eqn (1) will be awesome and highly desirable. Our point is that, with our goal to do “efficient” finding of disguised subnetworks, our current two-step relaxation is not only the most efficient; but also highly competitive in accuracy with no sacrifice.
>
> We thank you for your appreciation in our experiment design and results.

---

> ### Author Response · Authors · 2021-11-18
> **Response to Reviewer 7V6H (Part 2)**
>
> ## Numerical experiments
>
> You are correct that we are mainly comparing PaB with “without training methods”. However, showing this comparison is exactly our goal. Prior arts show mediocre accuracy with "NO" training on the weight values but still involve computationally heavy training to “train” the sparse structure. In contrast, we show *much better accuracy* without training the weight magnitudes too, but simply allow the weight to be pruned first and then extended with flipped signs. We believe what is demonstrated in this comparison is valuable. Moreover, how to implement the whole training process in a highly efficient way is also the second main goal and contribution of this work.
>
> We are aware of the theoretical efforts in (Malach et al. 2020) on top of (Ramanujan et al., 2019) and cited this work in the paper already. We came to awareness of the other two theoretical works (Pensia et al, 2020; Orseau et al., 2020) recently and will add them and the corresponding discussion about the theoretical results (as in our response to Reviewer QSy2) to the revised version. We believe that our PaB framework, as a more general extension to the “without training methods” category,  also enjoys the theoretical results mentioned. Our flexibility that allows sign flippings brings the possibility to improve the parameter complexity required in the larger network to achieve competitive accuracy.
>
> ## Minor Comments
>
> 1. *Page 2: “Even worse is..”.* We will rephrase it into “What is even worse is...”.
> 2. *Page 3: “Unfortunately,...”* We will rephrase it into “However,...”.
> 3. *Page 5: sparsity notation “s”* Thank you for pointing out the inconsistency in notations, for which we are very sorry. We have fixed it and will make sure of notation consistency in the final version.
> 4. *“Computationally infeasible” -> “computationally intractable”* We have rephrased this term as you suggested.
> 5. *“ = argmin “ -> “ \in argmin”* Thank you for this point. We have implemented this change.
> 6. *Page 5: “For example” connection* We will rephrase it into “That is, one sparse NN of k non-zero elements can be augmented to 2k possible candidates, if sign flipping is enabled.”
> 7. *Page 5: NP-hard terminology”* This is a good point. We will make our statement more accurate by rephrasing it into “computationally intractable to exhaustively search for the optimal vector”.
> 8. *Page 7: an extra comma* We have fixed this typo. Thank you!

---

> ### Author Response · Authors · 2021-11-28
> **Could you kindly read our response and share your thoughts?**
>
> Dear Reviewer 7V6H,
>
> We would like to thank you again for your detailed review. We wonder if you could kindly read through our response to your review and share your thoughts on it so that we can have the chance to address your further comments if there are any.
>
> In our response,
>
> 1. We tried to address your concerns about the contribution, novelty, and generality of the proposed PaB framework. We humbly argued that the two-step relaxation was a natural choice as efficient implementation was one of our major goals.
>
> 2. However, we are indeed aware that the two-step relaxation is not optimal. We conducted an iterative optimization experiment, which is closer to the joint optimization process. Obtaining no significant improvements, we think that the two-step relaxation is a reasonable choice that combines both efficiency and effectiveness.
>
> 3. We thank you for pointing out the minor issues in the original submission. We have uploaded a revised version that integrated all your comments.
>
> We would appreciate it if you could kindly take a look at both the revision and our response to your comments. If you have any further questions, we are happy to discuss them!
>
> Best,
>
> Authors

---

> > ### Comment · Reviewer_7V6H · 2021-11-30
> > **Authors' response is very clear, revision looks nice, but my main concern about theoretical contribution remains**
> >
> > I would like to thank the authors for their detailed answer, as well as for providing the revised document. In my view, all the points raised by the authors are clear, and the paper has very nice writing (as in the original submission).
> >
> > Regarding the training-free methods discussions: I appreciate the references given by the authors. The reason I asked this question in my official review is that I had thought the “catchy” part of the training-free methods was indeed them being training-free, which is a desired property in some areas (e.g., local privacy, distributed optimization, etc.). However, I don’t think this is an issue in the paper especially given the authors discuss that their method “enjoys the theoretical results [as in the training-free models, in terms of efficiency]”.
> >
> > My main concern about the structure of the optimization, however, remains. The authors implemented an iterative optimization process, and I would like to thank them for this. It is shown that this iterative solution does not improve the previous results significantly. I think this is understandable, since my previous point is still valid, which is: each iteration still first finds a hidden subnetwork, then converts it to a “disguised” one via sign flipping. I think here unless we consider different 'U' transformations, or actually solve a sub-problem where both decision variables are solved simultaneously, the results would not improve significantly (since the set of feasible solutions is extremely large and when we restrict ourselves to the sign flips, as well as the sequential solutions, we are restricting ourselves to a strictly progressive subset of the feasible set, hence an iterative adaptation of this method would not circumvent this issue).
> >
> > Still, when I say “unless we solve a sub-problem where both decision variables are solved simultaneously”, I am aware that there is no method that comes to mind immediately, and even if it did, I agree that this would change the positioning of this work. In my view, this work invites the community to do research focusing on problem (1), rather than being restricted to problem (2), by showing good empirical result. Hence, I would not be upset at all if this paper is accepted; while still having the concern that, despite having a very general definition of problem (2), the solution proposed comprises application of two known works, which is post-processing upon finding subnetworks.

---

> > > ### Author Response · Authors · 2021-11-30
> > > **Thanks for the reviewer's reply and futher discussion! Here are some of our thoughts**
> > >
> > > Dear Reviewer 7V6H,
> > >
> > > Thank you for your reply! We are happy that our previous response was clear to you and resolved part of your concerns.
> > >
> > > As for your major concern about the structure of the optimization, we agree with your point that it would be hard to see essential improvements unless we can truly optimize both decision variables in (1) simultaneously, i.e., the sparse mask and weight transformation. Exploring possible solutions that are at least closer to really addressing the joint optimization problem (1) will definitely be of interest to us in the future.
> > >
> > > However, we would like to add a few words about our solution to (1) in this work, i.e., the “post-processing upon finding subnetworks” as you mentioned, in our attempt to justify the technical contributions of this work.
> > >
> > > Results wise, yes, we adopted two known works --- we used SynFlow to find subnetworks and used Bop as post-processing to flip the signs. But we want to humbly emphasize that using Bop was not a trivial design choice here. And its empirical superiority also provided some insights on how the sparse masks could be optimized in a better way.
> > >
> > > Previous “training-free” methods [1,2,3] learn hidden scores for model parameters. The hidden scores decide how to mask parameters usually using some non-differentiable function, and thus can only be optimized with the straight-through-estimator (STE). We really wanted to get rid of hidden scores and STE due to the mediocre performance and zero benefits in reducing training cost.
> > >
> > > Therefore, we (for the first time) proposed to cast the “training-free” optimization as a BNN optimization problem and introduced Bop. Using Bop yielded better results as shown in our results (e.g., in Table 1) and naturally enabled an efficient training algorithm thanks to its hidden-score-free nature. The latter, as we emphasized in previous responses, is an important goal of this work.
> > >
> > > We had related discussion in our response to Reviewer QSy2, who expressed in the latest reply his/her appreciation in the non-straightforwardness of the utilization of Bop as one of our technical contributions.
> > >
> > > We hope this information will help to better justify the contribution of this work. We thank you again for all your valuable comments and suggestions on this work.
> > >
> > > Best,
> > >
> > > Authors
> > >
> > > ## References
> > >
> > > [1] Zhou, Hattie, et al. "Deconstructing lottery tickets: Zeros, signs, and the supermask." arXiv preprint arXiv:1905.01067 (2019).
> > >
> > > [2] Ramanujan, Vivek, et al. "What's Hidden in a Randomly Weighted Neural Network?." Proceedings of the IEEE/CVF Conference on Computer Vision and Pattern Recognition. 2020.
> > >
> > > [3] Ivan, Cristian, and Razvan Florian. "Training highly effective connectivities within neural networks with randomly initialized, fixed weights." arXiv preprint arXiv:2006.16627 (2020).

---

> > > > ### Comment · Reviewer_7V6H · 2021-11-30
> > > > **Replying positively to the authors**
> > > >
> > > > Dear Authors,
> > > >
> > > > I have previously restricted my review only to my personal view, which was equally close to accept/reject.
> > > >
> > > > However, having a look at all the reviews (and the follow-ups), I can see that two of the reviewers are very happy about the paper.  Reviewer rMZy, like me, gave a "5", with three reasons:
> > > > 1) The setting is new, but the practical value to the community seems unclear.
> > > > 2) The algorithm is straightforward, joint optimization is not considered.
> > > > 3) Experiments are promising but not exciting, ImageNet results are absent.
> > > >
> > > > I think the third point of Reviewer rMZy is fixed by the authors. The first point is, in my view, not an issue after the revision and I believe there is value in the work. The second point was my main concern.
> > > >
> > > > However, given that Reviewer QSy2 is also happy about the non-straightforwardness of the Bop stage, I would like to break the tie in favor of an acceptance.
> > > >
> > > > Thank you very much for the references provided, and your thorough answer. Congratulations on a nice paper.

---

> > > > > ### Author Response · Authors · 2021-11-30
> > > > > **Thank you for your active response!**
> > > > >
> > > > > Dear Reviewer 7V6H,
> > > > >
> > > > > Thank you for your active responses and the positive evaluation of our work post-rebuttal. The discussion was joyful. Thank you again for your valuable time!
> > > > >
> > > > > Best,
> > > > >
> > > > > Authors

---

### Official Review · Reviewer_QSy2 · 2021-11-02

**Correctness:** 4
**Technical Novelty And Significance:** 3
**Empirical Novelty And Significance:** 4
**Recommendation:** 8
**Confidence:** 4

**Main Review:**

Strengths:
The paper is well written, and does a fair job covering recent works.
The definition of disguised subnetworks is clear and properly motivated.
The presentation of the PaB is clear and well motivated with good context.
The results are quite convincing that PaB scaled well to larger networks,
something other methods lack, while providing good accuracy for much smaller
and easier to train models.

Weakness:

1. I would have liked to see some more theoretical discussion, though the authors
admit this work is primarily empirical.

2. It would be nice to discussion of other weight transformations U in the unmasking phase,
    that may be useful.

3. Though the definition of disguised subnetworks is new, and the generalized
approach is novel, and the results speak to the validity of the approach, the
PaB algorithm is a relatively straightforward application of two existing
methods.


**Summary Of The Paper:**

Optimizing sparse neural networks is an important topic due to their
computational and space savings.  Building on the work of the lottery ticket
hypothesis, others have shown there exist hidden subnetworks within randomly
initialized NN that have good performance. The authors extend this definition
to disguised subnetworks, which contain hidden subnetworks as a subclass.
Moreover, the authors present a novel combination of existing methods
into a single algorithm they call Peek-a-Boo (PaB) which can efficiently
find such networks.


**Summary Of The Review:**

The paper is well written, and does a good job covering all the related material.
The definition of disguised subnetworks is novel and will be useful for future researchers.
The PaB algorithm clearly yields good results on larger NN, something other methods lack.
More exploration of theoretical understanding as well as other weight transformations would be
useful.

---

> ### Author Response · Authors · 2021-11-18
> **Response to Reviewer QSy2**
>
> Dear Reviewer QSy2,
>
> Thank you for your positive feedback.
>
> ## More theoretical discussion
>
> Thank you for your suggestion. We are aware of several pieces of theoretical work about the capacity and expressibility of neural networks (NNs) with only sparse masks being optimized, such as (Malach et al., 2020; Pensia et al., 2020; Orseau et al., 2020). We cited the first one in Section 1 in the paper and will cite the other two as well in revision. All of these three theoretically prove that given any smaller network, we can find with high probability a larger randomly initialized network to approximate the smaller network as a function by simply optimizing the sparse mask and leaving the weights unchanged. The proposed PaB framework is more generalized because we allow further transformations on the remaining weights. This means that the results in the three theoretical works can also apply to PaB. A recent work on flipping-based training (Ivan & Florian, 2020), which has been cited in our manuscript, also provided interesting explanations that sign flipping can have similar effects in terms of minimizing cross entropy loss as pruning/masking.
>
> While it would be interesting to connect the above two parts of proof and to display the joint pruning-flipping benefit, we would like to humbly emphasize that this work is still mainly oriented towards empirical observations and improvements. We strive to find alternative ways to train sparse neural networks other than SGD and the efficient implementation of the proposed methods is also one of our major goals.
>
>
> ## Discussion about other weight transformations
>
> This is a very good point. While our work is mainly focused on sign flipping, there are several other options we can consider. For example, we can extend sign flipping with learnable scaling. One scaling factor will be learned for one layer and hence introduces near zero overheads in either storage or computation. Another example is quantization. We can quantize the remaining weights in PaB into several pre-defined values. We are happy to add this to the discussion section and explore in the future.
>
> ## Straightforward application of two existing methods?
>
> We would like to first thank you for your appreciation in the introduction of the concept of disguised networks, the proposed generalized framework and the good empirical results as our contributions. It may seem at first glance that the proposed algorithm is a combination of pruning-at-initialization methods and Bop training from BNN optimization. However, the selection of Bop is not straightforward. Before this work, the standard methods for training the sparse mask (or supermask in [2]) or the flipped signs use hidden weights and adopt STE. Although they work to some extent, we found that Bop type training is consistently better as shown in our results (e.g., in Table 1). The hidden-weight-free essence of Bop training also enables the potential of efficient training, which is also a very important and highlighted empirical contribution in this work. This observation of the limitation of the optimization with hidden weights is not trivial and is one important contribution of this work.
>
>
>
> ## References
>
> Pensia, Ankit, et al. "Optimal lottery tickets via subset sum: Logarithmic over-parameterization is sufficient." arXiv preprint arXiv:2006.07990 (2020).
>
> Orseau, Laurent, Marcus Hutter, and Omar Rivasplata. "Logarithmic pruning is all you need." NeurIPS, 2020.

---

> > ### Comment · Reviewer_QSy2 · 2021-11-29
> > **Thank you for clarifcations.**
> >
> > Thank you for the clarifications and references. Particularly on the non-straightforwardness on the selection of Bop training.

---

### Author Response · Authors · 2021-11-23
**A revised version of our paper is uploaded**

Dear all,

We thank you for the constructive comments and suggestions, which help us improve writing and clarify. We uploaded a revised version of the manuscript, in which we integrated a few revisions highlighted in blue color. The main edits include fixes to typos and the usage of language, additional references to the two theoretical work, a new discussion of alternative weight transformations in the discussion section and the results of the iterative optimization scheme suggested by Reviewer rMZy.

We would appreciate it if reviewers 7V6H and rMZy could kindly take a look at both the revision and our response to your comments to see if your concerns are resolved and your evaluation on our work is hopefully more positive. We thank everyone again for the valued efforts!

Best,

Authors

---

### Author Response · Authors · 2021-11-29
**Summary of Our Responses to Reviewers' Comments**

Dear reviewers and AC panel,

We would like to express our sincere gratitude for all your efforts put into the review process of our work and for the valuable and constructive comments from all reviewers. We appreciate that the merits of our work are recognized by Reviewer jduC and QSy2.

With only one day left before the end of the rebuttal period, we have not heard from the reviewers about their thoughts on our responses (after a general reply to everyone and a few mild individual reminders). Particularly, we are eager to hear from Reviewer 7V6H and rMZy because our responses to them contain essential information of the justification of our motivation, major goals, and the key algorithmic design. We summarize the major points we have emphasized during the rebuttal period so that you can grasp them quickly for a better understanding of our work:

1. **(Concern of contribution and novelty)** We argued that our contributions lied upon the two limitations of the edge-popup algorithm: (a) the strict constraint of optimizing masks only; and (b) the use of hidden scores and STE for training. Our solutions are (a) the new concept of “disguised networks”; and (b) the two-step relaxation for the joint optimization and utilization of the Bop algorithm. The new concept of “disguised networks” greatly enlarged the capacity of the original problem (supported by our results). The selection of Bop was not trivial and showed consistent improvements over edge-popup. Moreover, together with the two-step relaxation, it enabled hardware-efficient implementations of PaB.

2. **(Effectiveness of the two-step relaxation)** We emphasized that relaxing the joint optimization in eq. (1) to the two-phase process in eq. (3-4) was a natural choice for the algorithmic design if taking into account the efficient implementation of the PaB framework. And efficiency is a **major** and an important goal of this work. Meanwhile, we added the empirical results of the iterative optimization (suggested by Reviewer rMZy), which we found did not provide a clear improvement over the two-step version. While we agree with the reviewers that it would be better to directly solve the joint optimization problem (if possible, despite the intractable computation), PaB with the two-step relaxation is a good compromise that combines both efficiency and effectiveness. The additional results and discussion have been added to the Appendix in the revised version.

3. **(Related theoretical results)** We are aware of the theoretical results (Malach et al., 2020; Pensia et al., 2020; Orseau et al., 2020) on the expressibility of neural networks by optimizing sparse masks only. We humbly think that PaB is more general, thanks to the extension of sign flipping on the remaining weights; hence the theoretical results also apply to PaB. We have added those references to the revised version.

4. **(Scalability)** We provided the ImageNet results of PaB, which showed clear superiority over edge-popup.

Besides the major clarifications above, we also implemented fixes to the minor comments, such as rephrasing and more accurate math formulation. We are grateful for the reviewers pointing them out. We are eager for further discussions with the reviewers so that we can resolve your concerns. Thank you very much!

Best,

Authors

---

### Decision · Program_Chairs · 2022-01-20

**Decision:**

Accept (Poster)

**Comment:**

I recommend acceptance. This paper presents an interesting "in-between" of work on lottery tickets and work on supermasks, and I think it is sufficiently novel to merit acceptance even if the significance of the results will need to be left to the judgment of future researchers. The reviewers seem broadly in favor of acceptance, and I defer to their judgment as a proxy for that signal.

For a quick bit of context, work on "supermasks" (Zhou et al., 2019) has shown that randomly initialized networks contain subnetworks that can reach high accuracy without training the weights themselves. That is to say, within randomly initialized networks are high-accuracy subnetworks. This work is interesting in its own right and has had a number of interesting implications for the theoretical community. This work derives from work on the lottery ticket hypothesis (LTH; Frankle & Carbin 2019), which shows that randomly initialized networks contain subnetworks that can train to full accuracy on their own. The key distinctions between these two kinds of work are (1) the LTH trains the subnetworks, while supermask work does not and (2) the LTH work requires that the subnetworks train to full accuracy, while work on supermasks obtain high (but not full) accuracy in many cases. No one approach is "better" than the other; they simply showcase different properties of neural networks.

As far as I understand, this paper creates space for an "in-between:" high-accuracy subnetworks are created by finding subnetworks at random initialization and flipping the signs of some of the weights to improve accuracy. This is a limited modification to the subnetworks that falls short of actually training them (LTH work) but is more than leaving them at their random initializations (supermask work). Doing so appears to produce subnetworks that perform better than in supermask work but with a lighter-weight procedure than LTH work. The procedure for accomplishing this feat is different than either approach (using SynFlow to find the subnetwork and a binary neural network training scheme to find the signs), and there is probably significant room for improvement in this new algorithmic space (just as there was for both LTH and supermasks).

This is novel and interesting, and I defer to the reviewers who find it worthy of acceptance. I have reservations about the eventual significance of the work, but that determination will be made by future researchers.